# Ameliorative Effects of *Daphnopsis costaricensis* Extract against Oxazolone-Induced Atopic Dermatitis-like Lesions in BALB/c Mice

**DOI:** 10.3390/nu14214521

**Published:** 2022-10-27

**Authors:** Yunji Bae, Taeyoung Kim, Nojune Park, Sangho Choi, Dongkeun Yi, Silvia Soto, Nelson Zamora, Sunam Kim, Minhye Yang

**Affiliations:** 1College of Pharmacy, Pusan National University, Busan 46241, Korea; 2Natural Products Research Center, Korea Institute of Science and Technology, Gangneung 25451, Korea; 3International Biological Material Research Center, Korea Research Institute of Bioscience and Biotechnology, 111 Gwahak-ro, Yuseong-gu, Daejeon 34141, Korea; 4Bioprospecting Research Unit, National Biodiversity Institute, Heredia 22-3100, Costa Rica

**Keywords:** *Daphnopsis costariensis*, atopic dermatitis, anti-inflammatory, 7,8-dimethoxyflavone, 7,2’-dimethoxyflavone, interleukin 4, β-hexosaminidase

## Abstract

The genus *Daphnopsis* has been traditionally used as a purgative, diuretic, stimulant, and psoriasis treatment. In this study, the anti-AD (atopic dermatitis) activities of the *Daphnopsis costaricensis* EtOH extract (DCE) were investigated in an oxazolone (OX)-induced mouse model of AD, and the anti-inflammatory effects of its active compounds were confirmed in PI-sensitized or IgE/DNP-BSA-sensitized RBL-2H3 cells. DCE improved the symptoms of OX-induced inflammatory dermatitis (swelling, erythema, and increased ear thickening) in OX-induced BALB/c mice ears and reduced epidermal thickness and mast cell infiltration. Eleven flavonoid compounds were isolated from DCE, and two compounds (7,8-dimethoxyflavone and 7,2’-dimethoxyflavone) significantly inhibited IL-4 overexpression in PI-induced RBL-2H3 cells and mast cell degranulation in IgE + DNP-BSA-induced RBL-2H3 cells. Our study indicates that DCE and two compounds (7,8-dimethoxyflavone and 7,2’-dimethoxyflavone) might effectively improve inflammatory and atopic skin symptoms.

## 1. Introduction

Atopic dermatitis (AD) is a chronic skin disease caused by multiple factors. Although its main pathogenesis has not been clarified, it is known to be associated with genetic, immunological, environmental, and psychological factors [1]. Characteristic symptoms of AD are lichenification, dry skin, pruritus, eczema, and erythema [1], In particular, itching is the major diagnostic criterion of AD, and can have severe impacts on physical and emotional life [2]. Immunologically, activation of serum immunoglobulin E (IgE) production due to Th2 cytokine (IL-4 and IL-13) overexpression is the main cause of skin lesion formation [3] because it mediates inflammation by inducing the release of histamine and prostaglandin E from various inflammatory factors and mast cells [4,5]. In addition, it also damages the skin barrier, and, thus, facilitates the entry of inflammatory mediators into skin and repeats a vicious cycle of inflammation [6]. Currently, topical corticosteroids, topical calcineurin inhibitors, and antihistamines along with moisturizers are used to suppress these immune reactions in AD [7,8]. In addition, the monoclonal antibody dupilumab, which targets IL-4Rα and inhibits the biological actions of both IL-4 and IL-13, has recently been approved to treat adult patients with AD [9]. However, long-term treatment raises safety concerns of potential side effects (skin atrophy, increased percutaneous absorption, burning, and lethargy) [10,11].

Researchers are focusing on natural products to reduce the risks of conventional AD treatments [12], and various natural extracts have been reported to have the potential to treat AD [13,14]. Those described to date usually contain phenolics, such as flavonoids and coumarins, which are considered strong inhibitors of inflammatory and allergic reactions [15]. In particular, they potently inhibit the activities of inflammatory mediators such as IL-4, IgE, and proinflammatory cytokines in inflammatory animal models and cells [16,17]. Furthermore, many studies have demonstrated that natural products are safer and more effective compared to conventional chemical products [12,18]. Thus, natural products offer attractive alternatives for the treatment of AD.

*Daphnopsis costaricensis* is a small shrub of the family Thymelaeaceae indigenous to Costa Rica [19], and members of this genus have been used as diuretics, radical purgatives, and to treat psoriasis [20]. Previous phytochemical studies have identified secondary metabolites, including flavonoids, terpenoids, and terpenes, in this genus [20,21], but no such study has been conducted on *D. costaricensis*. Therefore, in the present study, to propose a potential natural anti-AD and anti-inflammatory substance, we investigated the effects of *D. costaricensis* EtOH extract (DCE) on AD-like lesions in an oxazolone (OX)-induced mouse model. In addition, we isolated and identified eleven compounds and examined their anti-inflammatory and anti-allergic activities on PI or IgE + DNP-BSA-induced RBL-2H3 cells.

## 2. Materials and Methods

### 2.1. Plant Material

The aerial parts of *Daphnopsis costaricensis* (Thymelaeaceae) were collected in the Osa Conservation Area, Golfo Dulce Forest Reserve, Costa Rica, in July 2013 and identified by Nelson Zamora (National Institute of Biodiversity, INBio). Voucher specimens (KRIB 0051617) are preserved at the International Biological Material Research Center (IBMRC) at the Korea Research Institute of Bioscience and Biotechnology, Daejeon, Republic of Korea.

### 2.2. General Experimental Procedures

JEOL 400 MHz (JEOL, Tokyo, Japan), Bruker 500 MHz (Bruker, Billerica, MA, USA), and Agilent Technologies 600 MHz instruments (Santa Clara, CA, USA) were used to obtain ^1^H, ^13^C, HMQC and HMBC NMR spectra. Sephadex LH-20 (25–100 μm; Pharmacia, Stockholm, Sweden) and silica gel (230–400 mesh; Merck, Darmstadt, Germany) were used to perform column chromatography, and Merck precoated silica gel 60 F_254_ Art. 5715 (Merck, Germany) plates were used to perform thin-layer chromatography (TLC). HR-ESI mass spectra were obtained using an Agilent Technologies 6530 Accurate-Mass Q-TOF LC/MS. Quantitative analysis was performed using a Shimadzu HPLC system (Tokyo, Japan) equipped with an SPD-20A UV/VIS detector, two LC-20AT pumps, and a CBM-20A HPLC system controller.

### 2.3. Extraction and Isolation

Air-dried aerial parts of *D. costaricensis* (4.4 kg) were subjected to three cycles of reflux extraction (95% EtOH (44 L) for 90 min at 30 °C and then held for 12 h at room temperature (RT)), filtered, and filtrates were combined and evaporated in vacuo at 40 °C to give DCE (225.3 g). The DCE obtained was suspended in distilled water (2 L) and partition extracted sequentially versus 4 L of *n*-hexane, ethyl acetate (EtOAc), and *n*-butanol (*n*-BuOH) to obtain 15.9 g, 13.2 g, and 32.3 g, respectively, of the corresponding extracts.

The EtOAc extract (13.2 g) was loaded to gradient silica gel column chromatography using Hexane:EtOAc (5:1 → 100% MeOH) as eluent to obtain 9 fractions (DCE1~DCE9). Fraction DCE4 (45 mg) was fractionated into three subfractions (DCE4-1~DCE4-3) using Sephadex LH-20 with MeOH. Subfraction DCE4-2 was subjected to RP HPLC (Watchers 120 ODS-BP, S-10 μm, 150 × 10 mm) at a flow rate of 2 mL/min using a UV detector (365 nm) and MeOH:H_2_O (80:20) as eluant to yield 6 subfractions (DCE4-2-1~DCE4-2-6), which afforded compound **1** (0.7 mg, *t*_R_ 15 min), and compound **2** (0.8 mg, *t*_R_ 20 min). Subfraction DCE4-2-5 was loaded to gradient silica gel column chromatography using Hexane:EtOAc (5:1 → 100% EtOAc) and afforded compounds **3** (1.9 mg) and **4** (1 mg). Fraction DCE6 (83.6 mg) was loaded to gradient silica gel column chromatography using Hexane:EtOAc (5:1 → 100% MeOH) to yield 11 fractions (DCE6-1~DCE6-11). Compound **5** (1.6 mg, *t*_R_ 40 min) and compound **6** (1.8 mg, *t*_R_ 55 min) were afford from subfraction DCE6-7 (18 mg) by RP HPLC (Watchers 120 ODS-BP, S-10 μm, 150 × 10 mm) using a flow rate of 2 mL/min, a 365 nm UV detector, and isocratic elution with MeOH:H_2_O (60:40) as eluent. Fraction DCE8 (577.1 mg) was fractionated into 12 fractions (DCE8-1~DCE8-12) using gradient silica gel column chromatography with Hexane:EtOAc (3:1 → 100% MeOH). Subfraction DCE8-7 (103.3 mg) was subjected to Sephadex LH-20 using MeOH to afford compound **7** (1.6 mg). Subfraction DCE8-9 (93.4 mg) was loaded to gradient silica gel column chromatography using Hexane:EtOAc (7:1 → 100% MeOH) to yield 8 fractions (DCE8-9-1~DCE8-9-8). DCE8-9-4 (16.1 mg) was conducted to RP HPLC (Watchers 120 ODS-BP, S-10 μm,150 × 10 mm) at a flow rate of 2 mL/min using a UV detector 330 nm by isocratic elution with 0.1% formic acid in ACN:0.1% formic acid in H_2_O (42:58) as eluant to afford compounds **8** (0.6 mg, *t*_R_ 40 min), **9** (1.5 mg, *t*_R_ 52 min), and **10** (0.8 mg, *t*_R_ 69 min). Subfraction DCE8-9-5 (23 mg) was conducted to RP HPLC (Watchers 120 ODS-BP, S-10 μm, 150 × 10 mm) using a flow rate of 2 mL/min, a 330 nm UV detector by isocratic elution with 0.1% formic acid in ACN:0.1% formic acid in H_2_O (36:64) to afford compound **11** (2.5 mg, *t*_R_ 31 min).

### 2.4. Animals

Female BALB/c mice of six-week-old were purchased from Orient Bio, Inc. (Seongnam, Korea), housed in a controlled environment (23 ± 3 °C and 55 ± 5% RH under a 12 h light/dark cycle), and allowed standard laboratory food and water ad libitum. All experimental procedures complied with the Guide for the Care and Use of Laboratory Animals issued by the National Institutes of Health (NIH publication No. 85-23, revised 2011) and were approved beforehand by the Institutional Animal Care and Use Committee of KIST (Certification No. KIST-2016-011).

### 2.5. Oxazolone-Induced Atopic Dermatitis BALB/c Mice

BALB/c mice were sensitized by applying 20 μL of 1% oxazolone in a mixture of acetone and olive oil (4:1) to ear surfaces once daily for 7 days. Subsequently, AD was induced by applying 20 μL of 0.1% oxazolone for 3 weeks every 2 days (the Oxazolone- group). During the AD induction period, mice in the DCE and Dexa groups were administered 1% DCE or 0.1% dexamethasone twice daily. Mice were divided into four groups (*n* = 4), that is, the OX, DCE, Dexa, and control (CON) groups. Animals in the CON group were administered distilled water instead of oxazolone throughout the sensitization and induction periods. On the final application day (day 28), mice were sacrificed, and samples were collected.

### 2.6. Histological Examination

Ear skins from BALB/c mice were fixed in 10% formaldehyde (Sigma) for 24 h, embedded in paraffin wax, serially sectioned at 4 μm, stained with hematoxylin and eosin (H&E) or toluidine blue for general morphology and mast cell infiltration. Epidermal thickness was measured using HKBasic software (KOPTIC, Seoul, Korea), and the number of mast cells infiltrating into the dermal layer was counted by randomly selecting three sections in toluidine blue-stained tissue. Histopathological changes (×200 magnification) were evaluated using the ProgRes^®^ CapturePro application software (JENOPTIK laser, Jena, Germany).

### 2.7. Cell Culture

RBL-2H3 (a rat basophilic leukemia cell line) cells were purchased from the American Type Culture Collection (ATCC, Rockville, MD, USA). RBL-2H3 cell were maintained in 150 cm^2^ cell culture dish with DMEM (Dulbecco’s modified essential medium, HyClone, Logan, UT, USA) containing 10% FBS (fetal bovine serum), 100 U/mL penicillin, and 100 μg/mL streptomycin (HyClone) at 37 °C in a humidified atmosphere containing 5% CO_2_.

### 2.8. Measurement of IL-4 mRNA Expression

RBL-2H3 cells were injected with DMSO or compounds isolated from DCE (10 μM) for 1 h, and inflammation was triggered by adding PI (PMA (phorbol 12-myristate 13-acetate, Sigma-Aldrich, St. Louis, MO, USA) at 50 ng/mL and ionomycin at 1 μM (Sigma-Aldrich), which induced states comparable to AD [22]. The control group was injected with DMSO without PI. After treatment for 20 h, total mRNA was harvested to synthesize cDNA, and IL-4 mRNA levels were calculated by quantitative real-time PCR (qPCR). Total RNA extraction was accomplished with the RNeasy mini kit (Qiagen, Valencia, CA, USA), and cDNA synthesis was conducted with the RevertAid First Strand cDNA Synthesis Kit (Thermo Fisher, Waltham, MA, USA). qPCR was performed using the QuantStudio™ 6 Pro Real-Time PCR System (Applied Biosystems, Foster City, MA, USA) and Fast SYBR^®^ Green Master Mix (Applied Biosystems). qPCR analysis was repeated twice and designed by duplication per sample. Expression levels of cytokines in exposed cells were compared to those in control cells at predetermined time points using the comparative cycle threshold (Ct) method. The sequences of the primers used were IL-4 forward: 5′-ACC TTG CTG TCA CCC TGT TC-3′; IL-4 reverse: 5′-TTG TGA GCG TGG ACTCAT TC-3′; β-actin forward: 5′-TCA TCA CCA TCG GCA ACG-3′, β-actin reverse: 5′-TTC CT GAT GTC CAC GTC GC-3′. mRNA expressions were normalized versus β-actin.

### 2.9. Measurement of β-Hexosaminidase Release

RBL-2H3 cells were seeded in 24-well plates, cultured overnight, sensitized with 100 ng/mL IgE for 4 h at 37 °C in 5% CO_2_. The IgE-sensitized cells were incubated with each of the eleven compounds (10 μM) isolated from DCE for 1 h, followed by 30 min incubation with 10 μg/mL DNP-BSA to stimulate degranulation. To assess β-hexosaminidase activity, the culture medium was supplemented with 10 mM poly-N-acetyl glucosamine in 0.1 M sodium citrate buffer (pH 4.5) in 96-well plates and incubated for 1 h at 37 °C. Absorbances at 405 nm were read using a microplate reader (Tecan Infinite M1000 Microplate Reader, Männedorf, Switzerland) [23].

### 2.10. Statistical Analysis

In vitro data are shown as means ± standard deviations (SDs) and in vivo data as mean ± standard errors of means (SEMs) (*n* = 4). Statistical analysis was performed using one-way analysis of variance (ANOVA). *p* values of < 0.05 were considered statistically significant.

## 3. Results

### 3.1. Effects of D. costaricensis EtOH Extract (DCE) on Oxazolone-Induced AD-Like Lesions 

To verify the anti-AD effect of DCE on mouse ear skin, BALB/c mice were administered OX for four weeks. OX-induced mice showed AD-like skin symptoms, which included increases in the ear and epidermal thickness, tumefaction, erythema, and dry skin, but it was improved in the DCE group (Figure 1A). On the final application day (day 28), the animals were sacrificed and ear thickness was determined. Ear thickness on the last day of the experiment was thicker in the OX group (0.32 mm) than in the CON group (0.19 mm) (Figure 1B). However, DCE significantly reduced ear thickness (to 0.24 mm) versus the OX group. The Dexa group (0.16 mm), used as positive control, also showed reduced ear thickness versus the OX group.

### 3.2. Histopathologic Effects of DCE on Oxazolone-Induced AD-Like Lesions

To examine the anti-AD effect of DCE on histopathological features, the epidermal tissues of mouse ears were stained with H&E or toluidine blue to determine epidermal thicknesses and degrees of mast cell infiltration. Microscopic examination revealed epidermal thickness was thinner in the DCE group than in the OX group (Figure 2A). Mast cell counts were also lower in the DCE group (Figure 3A). Oxazolone treatment increased epidermal thickness and mast cell infiltration by 2- and 4.5-fold, respectively, versus the CON group (Figure 2B and Figure 3B). However, in the DCE group, epidermal thickness was reduced by 22%, and mast cell infiltration was significantly reduced by 45% versus the OX group. In the Dexa group, the epidermal thickness and mast cell infiltration were reduced by 65% and 63%, respectively, versus the OX group.

### 3.3. Isolation of Compounds from DCE

Figure 4 provides details of the eleven compounds isolated from DCE as identified by 1D and 2D NMR and HR-MS data, and comparisons with the literature. The isolated compounds were identified as follows: 5,2′-dimethoxyflavone (**1**) [24], 6-methoxyflavone (**2**) [24], 6′-methoxyflavone (**3**) [25], 7-methoxyflavone (**4**) [24], 7,8-dimethoxyflavone (**5**) [26], 7,2′-dimethoxyflavone (**6**) [27], 7-hydroxyflavone (**7**) [28], 6,7-dimethoxyflavone (**8**) [29], 7,8,6′-trimethoxyflavone (**9**) [30], 5,7,8,2′-tetramethoxyflavone (**10**) [31], and 7-hydroxy-6′-methoxyflavone (**11**) [32].

The active compound **5** exhibits a peak at *m*/*z* 283.0974 [M+H]^+^ in HREISMS, corresponding to its molecular formula as C_17_H_14_O_4_. The ^1^H NMR spectrum of **5** indicated five aromatic protons of B-ring at δ_H_ 8.08 (m, 2H, H-2′, H-6′), 7.61 (m, 3H, H-3′, H-4′, H-5′), and *ortho*-coupling protons of A-ring at δ_H_ 7.79 (d, *J* = 8.9 Hz, 1H, H-5), 7.29 (d, *J* = 8.9 Hz, 1H, H-6). The observation of a sharp singlet at δ_H_ 6.98 (s, 1H) with the above data and a carbonyl group at δc 176.6 (C-4) showed that compound **5** was a flavone. In addition, the positions of two methoxy groups were determined by correlations between δ_H_ 3.96 (d, 6H, OCH_3_-7, OCH_3_-8) and δc 156.5 (C-7), 136.4 (C-8) in the HMBC spectrum. Based on this spectral data, the structure of **5** was identified as 7,8-dimethoxyflavone. 

The active compound **6** has a molecular formula C_17_H_14_O_4_ as a peak at *m*/*z* 283.0968 [M+H]^+^ in HREISMS. The ^1^H NMR spectrum of **6** identified 1′, 2′-substituted aromatic protons of B-ring at δ_H_ 7.94 (m, 1H, H-6′), 7.58 (td, J = 8.49, 7.4, 1.74 Hz, 1H, H-4′), 7.26 (m, 1H, H-3′), 7.16 (td, J = 7.4, 7.2, 2.4 Hz, 1H, H-5′), and *ortho*-coupling protons [δ_H_ 7.94 (d, J = 8.0 Hz, 1H, H-5), 7.07 (dd, J = 8.8, 2.3 Hz, 1H, H-6)] and a singlet [δ_H_ 7.26 (s, 1H, H-8)] of A-ring. Moreover, a singlet at δ_H_ 6.86 (s, 1H, H-3) and a carbonyl group at δc 176.4 (C-4) proved that compound **6** was a flavone. The positions of two methoxy groups were determined by correlations between δ_H_ 3.93 (s, 3H, OCH_3_-2′) and δc 157.5 (C-2′), and δ_H_ 3.91 (s, 3H, OCH_3_-7) and δc 163.8 (C-7), respectively, in the HMBC spectrum. On the basis of these spectral data, the structure of **6** was determined as 7,2′-dimethoxyflavone.

### 3.4. Compounds Isolated from DCE and Their Effects on RBL-2H3 Cells

The anti-inflammatory and anti-allergic effects of the eleven compounds isolated from DCE were assessed using IL-4 levels and β-hexosaminidase degranulation. PI-stimulation significantly increased the gene expression of IL-4 in RBL-2H3 cells. Pretreatment with 7,8-dimethoxyflavone (**5**), 7,2′-dimethoxyflavone (**6**), or 6,7-dimethoxyflavone (**8**) inhibited PI-induced increases in IL-4 levels by 46%, 35%, and 35%, respectively (Figure 5A). Furthermore, DNP-BSA-stimulation induced the release of β-hexosaminidase in IgE-sensitized RBL-2H3 cells, and 6′-methoxyflavone (**3**), 7-methoxyflavone (**4**), 7,8-dimethoxyflavone (**5)**, or 7,2′-dimethoxyflavone (**6**) pretreatment inhibited β-hexosaminidase release by 17%, 16%, 27%, and 30%, respectively, versus IgE + DNP-BSA-stimulated RBL-2H3 cells (Figure 5B).

## 4. Discussion

Flavonoids are extensively produced by plants and present in edible plants, fruits, legumes, and tea [33]. The number of studies performed on plant flavonoids is increasing as they have been shown to have beneficial protective effects on human health without any side effects [34]. Structurally, plant-derived flavonoids have a C6-C3-C6 carbon skeleton with different substitution patterns, as demonstrated by luteolin, apigenin, diosmetin, and quercetin [34,35]. Plant flavonoids have been shown to act as anti-inflammatory agents in epidemiologic, clinical, and animal studies [35,36]. In particular, they have been shown to regulate various inflammation-related enzyme systems and transcription factors [37], and some have been reported to have affirmative therapeutic effects on chronic inflammatory skin diseases (atopic dermatitis, urticaria, and psoriasis) [37,38]. Accordingly, the discovery of novel flavonoids is attracting attention as a means of identifying prospective candidate anti-inflammatory drugs.

The Thymelaeaceae family has a broad range of biological activities, which include anti-inflammatory, anti-cancer, and antibacterial activities, and has been used to treat human diseases for centuries [20,21]. Plant extracts of the Thymelaeceae family commonly contain highly diverse phenolics and possess potent anti-inflammatory properties [21]. Representatively, *Daphne* species inhibited acute and chronic inflammation in an inflammation-induced mouse model [39,40]. *Wikstroemia* species ameliorated AD-like skin lesions and decreased serum IL-4 and IgE levels in mice [41], and the genus *Daphnopsis* has been used in diuretic, laxative, and psoriasis remedies [20]. Although bioactive compounds, including flavonoids and terpenoids, with anti-inflammatory activities have been reported in the genus *Daphnopsis*, studies on the treatment of inflammatory diseases in this genus are still rare. For this reason, we undertook to characterize the anti-AD effect of DCE in an AD-induced mouse model and to examine the anti-allergic and anti-inflammatory effects of bioactive compounds isolated from DCE on inflammatory mediator degranulation in vitro.

AD is a representative allergic inflammatory skin disease. Patients with allergic inflammatory skin diseases typically exhibit symptoms such as keratinization, swelling, and erythema [42]. These inflammatory hypersensitivity reactions increase mast cell infiltration in the epidermis and thicken the epidermis [43]. Anti-AD examination of DCE via a representative oxazolone-induced BALB/c mouse model has been performed, and mice with repeated application of oxazolone exhibited erythema and edematous dermatitis along with immunological inflammatory response [44]. As a result, DCE improved AD-like lesions, severe scratching behavior, and ear thickness induced by oxazolone treatment. In the histological study, H&E and toluidine blue staining showed DCE significantly reduced mast cell infiltration and epidermal thickening. These results show that DCE has the potential to suppress allergic inflammatory symptoms in skin.

Inflammatory hypersensitivity reactions promote the differentiation of T cells and the release of various inflammatory cytokines, including IL-4, by Th2 cells [43]. Excessive IL-4 secretion degranulates immune cells, exacerbates epidermal barrier dysfunction, and causes itching [6,45]. For this reason, many authors considered IL-4 inhibitors key markers for the development of anti-inflammatory and allergy treatments [46]. Therefore, we examined the expression of mRNA IL-4 and degranulation of mast cells in PI or IgE/DNP-BSA pretreated RBL-2H3 cells subsequently treated with each of the eleven isolated flavonoids from DCE. RBL-2H3 cells have commonly been reported to be a mucosal mast cell line, and extensively used to study IgE–FcεRI interactions [47,48]. We found that 7,8-dimethoxyflavone (**5**), 7,2′-dimethoxyflavone (**6**), and 6,7-dimethoxyflavone (**8**) significantly decreased the expression of IL-4 mRNA. In addition, 6’-methoxyflavone (**3**), 7-methoxyflavone (**4**), 7,8-dimethoxyflavone (**5**), and 7,2′-dimethoxyflavone (**6**) showed strong inhibitory effects in the β-hexosaminidase release assay which is considered as a biomarker of mast cell degranulation. These observations support previous studies that methoxylation of the flavone A-ring 7-position enhances the anti-inflammatory activity and that multiple substituents have little effect on anti-inflammatory activity [49]. In addition, our results support earlier studies that hydroxylation of the flavone A-ring-7 position attenuates the anti-inflammatory activity, while hydroxyl groups on the flavone B-ring increase their inhibitory action [50]. Our results suggest that DCE containing flavonoids with anti-allergic and anti-inflammatory effects might be useful for the development of prophylactic and therapeutic agents for AD.

## 5. Conclusions

In summary, this study shows that DCE ameliorated AD-like pathology by decreasing ear epidermal thicknesses and mast cell infiltration in an oxazolone-induced BALB/c mouse model. Eleven active compounds were isolated from DCE, and 7,8-dimethoxyflavone and 7,2′-dimethoxyflavone were found to inhibit IL-4 overproduction and mast cell degranulation in vitro. Accordingly, our results provide that DCE has potential use as a natural treatment for AD and chronic skin disease.

## Figures and Tables

**Figure 1 nutrients-14-04521-f001:**
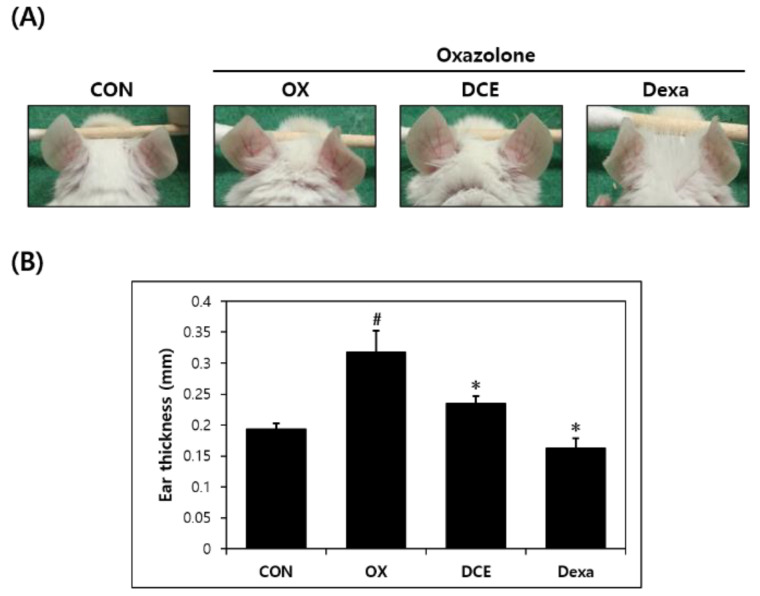
Effect *D. costaricensis* EtOH extract (DCE) on OX-induced AD-like lesions in the BALB/c mouse model. (**A**) Clinical features of AD-like symptoms. (**B**) Ear thickness. CON group: vehicle controls, OX group: oxazolone-treated controls, DCE group: oxazolone plus 1% DCE-treated, and Dexa group: oxazolone plus 0.1% dexamethasone-treated. Data values are presented as means ± SEMs. ^#^
*p* < 0.05 vs. the CON group, * *p* < 0.05 vs. the OX group.

**Figure 2 nutrients-14-04521-f002:**
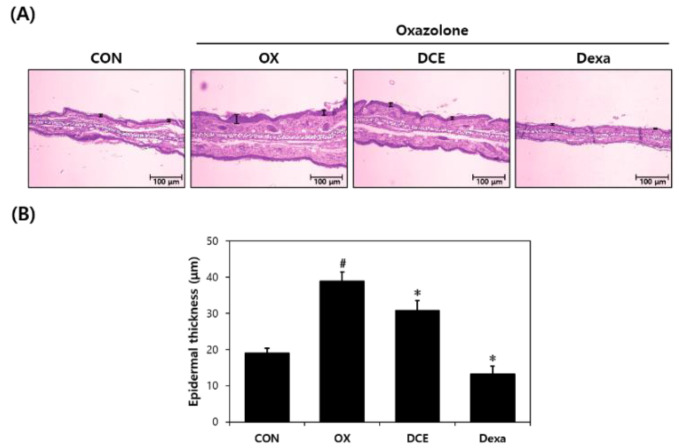
Effect of *D. costaricensis* EtOH extract (DCE) on H&E findings and epidermal thickness. (**A**) H&E staining results. (**B**) Epidermal thicknesses of mouse ears. CON group: vehicle controls, OX group: oxazolone-treated controls, DCE group: oxazolone plus 1% DCE-treated, and Dexa group: oxazolone plus 0.1% dexamethasone-treated. Data values are expressed as the means ± SEMs. ^#^
*p* < 0.05 vs. the CON group, * *p* < 0.05 vs. the OX group.

**Figure 3 nutrients-14-04521-f003:**
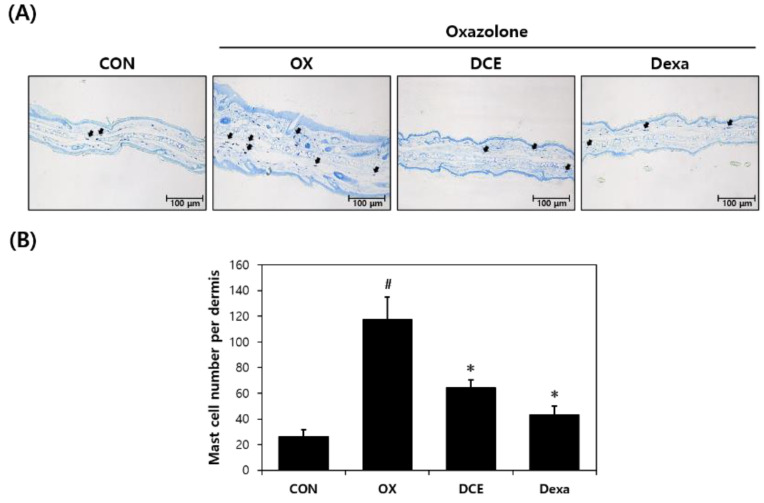
Effect of *D. costaricensis* EtOH extract (DCE) on toluidine blue staining findings and mast cell numbers in dermal tissues. (**A**) Toluidine blue staining results. (**B**) Mast cell numbers in dermal tissue. CON group: vehicle controls, OX group: oxazolone-treated controls, DCE group: oxazolone plus 1% DCE-treated, and Dexa group: oxazolone plus 0.1% dexamethasone-treated. Data values are expressed as means ± SEMs. ^#^
*p* < 0.05 vs. the CON group, * *p* < 0.05 vs. the OX group.

**Figure 4 nutrients-14-04521-f004:**
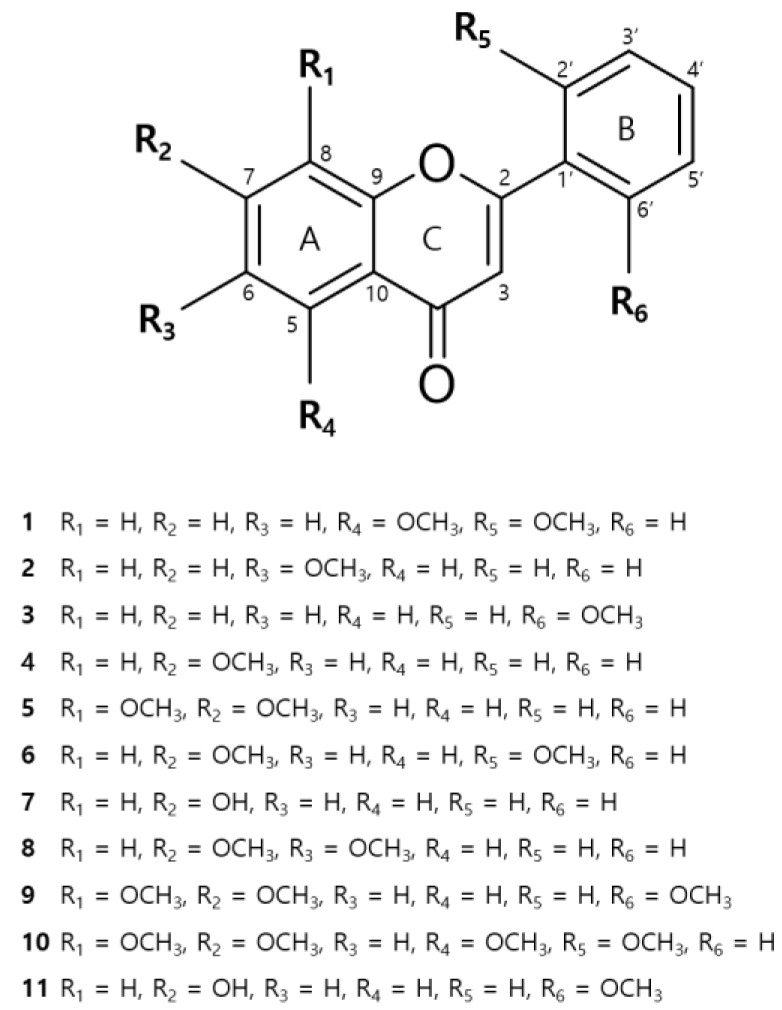
Chemical structures of the compounds isolated from DCE.

**Figure 5 nutrients-14-04521-f005:**
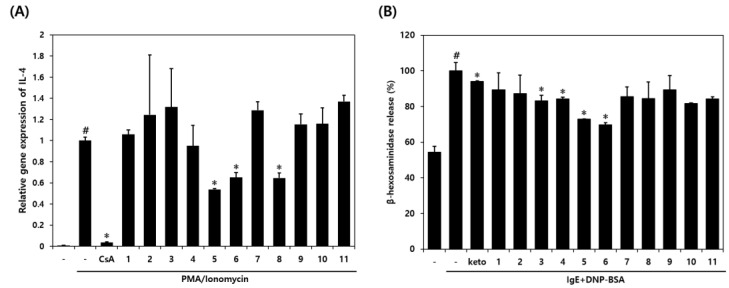
Anti-inflammatory and anti-allergic effects of compounds isolated from DCE on IL-4 levels and β-hexosaminidase release. (**A**) Expressions of IL-4. (**B**) β-hexosaminidase release. CsA: PI plus cyclosporin A-treated group, keto: IgE + DNP-BSA plus ketotifen-treated group. Data values are presented as the means ± SDs. ^#^
*p* < 0.05 vs. vehicle controls, * *p* < 0.05 vs. the PI or IgE + DNP-BSA-stimulated cells.

## Data Availability

Data are contained within the article.

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
