# Peer review of "Ameliorative Effects of Daphnopsis costaricensis Extract against Oxazolone-Induced Atopic Dermatitis-like Lesions in BALB/c Mice"

_nutrients, 2022, doi:10.3390/nu14214521_

Round 1
Reviewer 1 Report
The authors investigated the potency of a number of fractions of an extract from Daphnopsis costaricensis against induced AD in mice and the inflammatory response of RBL-2H3 cells.
1. IL-13 plays also a key role in AD, as it activates as well the type II IL-4 receptor complex. A discussion about this would complete the picture and in lines 41-44 authors induce the idea that IL-4 is the main cause of the installation of the symptoms of AD. This should be corrected, or nuanced, after the case.
2. Lines 155-157 need references.
3. Oligos used for gene expression of Il-4 does not align with the sequence NR_027491.1, nor NM_021283.2 (IL-4 precursor), and they seem to anneal with a variety of other mRNA sequences from the mouse genome. Could you please provide details on the way the primers were selected?
4. Lines 170-176 need citation
5. Pictures in Figure 2A and Figure 3A should have better resolutions; please include a micrometric scale.
6. Please provide methodological details for the results presented in Figure 3B
7. For chapter 3.3 please provide description of the compounds identification according details provided in chapter 2.3.
8. The reader would benefit from a dissertation on the rationale behind chosing RBL-2H3 cell line for this study.
9. Lines 288-289: the stipulated significant decreased IL-4 mRNA levels by DCE 5,6,8 is not sustained by the results presented in Figure 5A. A fold decrease of <0.4 is not considered relevant for e gene expression study. Considering the limitations of the SYBR Green RT-PCR method and the concerns related to oligos design, the results fail to convince that described compounds really produce an amelioration of the inflammatory state.
10. How many animals were used in the study?
11. Please provide details on the RT-PCR method applied for gene expression (replicates, number of samples, interplate calibrators)
Author Response
Reviewer #1
- IL-13 plays also a key role in AD, as it activates as well the type II IL-4 receptor complex. A discussion about this would complete the picture and in lines 41-44 authors induce the idea that IL-4 is the main cause of the installation of the symptoms of AD. This should be corrected, or nuanced, after the case.
- We appreciate reviewer’s valuable comments. The importance of IL-13 in AD is now newly added such as ‘Immunologically, activation of serum immunoglobulin E (IgE) production due to Th2 cytokines (IL-4 and IL-13) overexpression is the main cause of skin lesion formation [3].’ and ‘In addition, the monoclonal antibody dupilumab, which targets IL-4Rα and inhibits the biological actions of both IL-4 and IL-13, has recently been approved to treat adult patients with AD [9].’.Please see yellow-highlighted parts in Introduction (lines 41-43/48-50).
- Lines 155-157 need references.
- It is now newly added such as Ref 22.
- Oligos used for gene expression of Il-4 does not align with the sequence NR_027491.1, nor NM_021283.2 (IL-4 precursor), and they seem to anneal with a variety of other mRNA sequences from the mouse genome. Could you please provide details on the way the primers were selected?
- The cells used in Figure 5A are RBL-2H3 cells that are basophilic cell lines isolated from rats, not mice. Rattus norvegicus mRNA for interleukin 4 (GenBank: NM_201270.1). Primers were designed from NCBI Primer-BLAST.
- Lines 170-176 need citation
- It is now newly added such as Ref 23.
- Pictures in Figure 2A and Figure 3A should have better resolutions; please include a micrometric scale.
- The picture resolution of Figure 2A and Figure 3A is now improved, and a scale bar is now newly added.
- Please provide methodological details for the results presented in Figure 3B
- Details for the methods about epidermal thickness and mast cell counting procedures are now newly added to Chapter 2.6 (lines 145-148), such as ‘Epidermal thickness was measured using HKBasic software (KOPTIC, Seoul, Korea), and the number of mast cells infiltrating into the dermal layer was counted by randomly selecting three sections in toluidine blue-stained tissue.’.
- For chapter 3.3 please provide description of the compounds identification according details provided in chapter 2.3.
- Description of two active compounds identification is now newly added to the Results, such as
‘The active compound 5 exhibits a peak at m/z 283.0974 [M+H]+ in HREISMS, corresponding to its molecular formula as C17H14O4. The 1H NMR spectrum of 5 indicated five aromatic protons of B-ring at δH 8.08 (m, 2H, H-2’, H-6’), 7.61 (m, 3H, H-3’, H-4’, H-5’), and ortho-coupling protons of A-ring at δH 7.79 (d, J = 8.9 Hz, 1H, H-5), 7.29 (d, J = 8.9 Hz, 1H, H-6). The observation of a sharp singlet at δH 6.98 (s, 1H) with the above data and a carbonyl group at δC 176.6 (C-4) showed that compound 5 was a flavone. In addition, the positions of two methoxy groups were determined by correlations between δH 3.96 (d, 6H, OCH3-7, OCH3-8) and δC 156.5 (C-7), 136.4 (C-8) in the HMBC spectrum. Based on this spectral data, the structure of 5 was identified as 7,8-dimethoxyflavone.
The active compound 6 has a molecular formula C17H14O4 as a peak at m/z 282 [M+H]+ in HREISMS. The 1H NMR spectrum of 6 identified 1’, 2’-substituted aromatic pro-tons of B-ring at δH 7.94 (m, 1H, H-6’), 7.58 (td, J = 8.49, 7.4, 1.74 Hz, 1H, H-4’), 7.26 (m, 1H, H-3’), 7.16 (td, J = 7.4, 7.2, 2.4 Hz, 1H, H-5’), and ortho-coupling protons [δH 7.94 (d, J = 8.0 Hz, 1H, H-5), 7.07 (dd, J = 8.8, 2.3 Hz, 1H, H-6)] and a singlet [δH 7.26 (s, 1H, H-8)] of A-ring. Moreover, a singlet at δH 6.86 (s, 1H, H-3) and a carbonyl group at δc 176.4 (C-4) proved that compound 6 was a flavone. The positions of two methoxy groups were determined by correlations between δH 3.93 (s, 3H, OCH3-2’) and δC 157.5 (C-2’), and δH 3.91 (s, 3H, OCH3-7) and δC 163.8 (C-7), respectively, in the HMBC spectrum. On the basis of these spectral data, the structure of 6 was determined as 7,2'-dimethoxyflavone.’. Please see yellow highlighted parts (lines 237-255).
- The reader would benefit from a dissertation on the rationale behind chosing RBL-2H3 cell line for this study.
- RBL-2H3 cells are similar to mucosal mast cells, and RBL2H3 cells have IgE receptors on the cell surface and secrete immune cytokines. Therefore, RBL2H3 cells are used to study IgE-dependent anti-allergic responses. We now newly added the above information and references (Ref 47,48) to the Discussion part. Please see yellow-highlighted part (lines 320-321).
- Lines 288-289: the stipulated significant decreased IL-4 mRNA levels by DCE 5,6,8 is not sustained by the results presented in Figure 5A. A fold decrease of <0.4 is not considered relevant for e gene expression study. Considering the limitations of the SYBR Green RT-PCR method and the concerns related to oligos design, the results fail to convince that described compounds really produce an amelioration of the inflammatory state.
- The IL-4 mRNA levels of DCE5, 6, 8 shown in Figure 5A is now corrected. In Figure 5A, the mRNA levels of DCE 5, 6, 8 represent a 40% decrease versus PMA+Ionomycin (PI) induction group. As a result of calculating the delta delta Ct value, the PI induction group increased 122 times versus the untreated group, and DCE 5, 6, and 8 compounds-treated increased 65, 79, and 70-fold, indicating a significant result. In addition, when gel loading using the qPCR products, a single band was shown and it was confirmed that amplification was performed using an appropriate primer.
- How many animals were used in the study?
- The study was conducted in four groups of four animals per group. Please see yellow-highlighted part (lines 137-138).
- Please provide details on the RT-PCR method applied for gene expression (replicates, number of samples, interplate calibrators)
- qPCR analysis was repeated twice, and designed by duplication per sample. An interplate calibrator was not used as the experiments were conducted using one plate. Detail for the RT-PCR method are now newly added to Chapter 2.8 (line 168).
Reviewer 2 Report
Jun-Ji Bae and associates in their manuscript describe data from investigation on the effect of Daphnopsis castaricensis extract on progression of atopic dermatitis-like lesions induced by administration of oxazolone to ear surfaces of BALB/c mice.
In general, the experiments were well planned and executed. Overall, the studies are described clearly and although, the reported results are confirmatory to some extent, they also provide information that might be useful for development of more effective therapy in chronic skin disease.
However, some minor corrections are necessary. The CsA and Keto abbreviation should be described in Fig. 5 legend.
Author Response
- We appreciate reviewer’s valuable comment. It is now newly added to Figure 5, such as ‘CsA: PI plus cyclosporin A-treated group, keto: IgE + DNP-BSA plus ketotifen-treated group.’. Please see yellow highlighted part (lines 271-272).
Round 2
Reviewer 1 Report
The authors addressed most of the notes made in the first round of review.